# Deep learning-based assessment of pulp involvement in primary molars using YOLO v8

**Aydin Sohrabi**[1,2], **Nazila Ameli**[2], **Masoud Mirimoghaddam**[2], **Yuli Berlin-Broner**[2], **Hollis Lai**[2], **Maryam Amin**[2]*

**1** Department of Orthodontics, Faculty of Dentistry, Tabriz University of Medical Sciences, Tabriz, Iran,
**2** Faculty of Medicine & Dentistry, Mike Petryk School of Dentistry, University of Alberta, Edmonton, Canada

* maryam.amin@ualberta.ca

## Abstract

Dental caries is a major global public health problem, especially among young children. Rapid decay progression often necessitates pulp treatment, making accurate pulp condition assessment crucial. Despite advances in pulp management techniques, diagnostic methods for assessing pulp involvement have not significantly improved. This study aimed to develop a machine learning (ML) model to diagnose pulp involvement using radiographs of carious primary molars. Clinical charts and bitewing radiographs of 900 children treated from 2018-2022 at the University of Alberta dental clinic were reviewed, yielding a sample of 482 teeth. images were preprocessed, standardized, and labeled based on clinical diagnoses. Data were split into training, validation, and test sets, with data augmentation applied to classify 2 categories of outcomes. The YOLOv8m-cls model architecture included convolutional and classification layers, and performance was evaluated using top-1 and top-5 accuracy metrics. The YOLOv8m-cls model achieved a top-1 accuracy of 78.7% for upper primary molars and 87.8% for lower primary molars. Validation datasets showed higher accuracy for lower primary teeth. Performance on new test images demonstrated precision, recall, accuracy, and F1-scores, highlighting the model's effectiveness in diagnosing pulp involvement, with lower primary molars showing superior results. This study developed a promising CNN model for diagnosing pulp involvement in primary teeth using bitewing radiographs, showing promise for clinical application in pediatric dentistry. Future research should explore whole bitewing images, include clinical variables, and integrate heat maps to enhance the model. This tool could streamline clinical practice, improve informed consent, and assist in dental student training.

## Author summary

We developed an AI-based diagnostic tool for identifying dental pulp involvement in primary teeth, which is a common complication of severe cavities requiring careful assessment to determine optimal treatment plans. Our system analyzes dental X-rays of

**Data availability statement:** The data supporting this study's findings are available from the University of Alberta. However, restrictions apply to the availability of these data, which were used under license for the current study and are not publicly available. The data can be requested via email at dentrsch@ualberta.ca, subject to request and with permission from the University of Alberta.

**Funding:** This study was supported by the University of Alberta, School of Dentistry, Endodontics Endowment Fund (Fund # E0162). The funder had no role in the study design, data collection, analysis, interpretation, or publication decision. The corresponding author is the Alberta Dental Association and College Clinical Dentistry Research Chair.

**Competing interests:** The authors have declared that no competing interests exist.

primary teeth to assist in diagnosis of pulp involvement in deep caries. Results demonstrate high diagnostic accuracy, offering several benefits: faster, more reliable diagnoses improving treatment efficiency; enhanced communication between dental practitioners and parents; and educational value as a training resource for dental students to supplement their limited clinical experience and improve their learning. By using modern technology in children's dental care, we hope to make diagnoses more accurate and ultimately enhance children's oral health outcomes.

## Introduction

Dental caries is a serious global public health problem, especially among young children. The severity ranges from the white spot lesions in enamel to extensive cavitation [1]. According to the National Health and Nutrition Examination Survey, 51% of children aged 6–8 in the United States have dental caries in their primary teeth [2].

Caries in the primary teeth tend to progress rapidly and extend to involve the pulp that necessitates pulp treatment [3]. Therefore, assessing the pulp condition is a crucial diagnostic procedure in pediatric dentistry for reaching a proper treatment decision. Conventionally, dentists rely on patients' history of pain, clinical examinations, and radiographic images of the teeth and the periodontal tissues, as well as pulpal and periodontal diagnostic tests performed [4]. The three techniques commonly used to manage pulpitis in primary teeth include direct pulp capping, pulpotomy, and pulpectomy; each using various evolving and improving materials. These techniques have been successfully developed over the past few decades. However, there has not been a significant advancement in the diagnostic methods for determining whether dental caries have reached the pulpal tissue.

One of the most challenging decisions in the treatment of carious primary molars is assessing the severity and progression of caries and accurately determining if the decay has reached the pulp, as well as the extent of pulp inflammation or infection. The reliability of the dental history provided by the child patient complicates clinical decisions about the probability of pulp involvement and the need for pulp treatment. Clinicians, therefore, commonly make these decisions relying on bitewing images to assess the location and extent of caries, as well as any evidence of radicular pathology in x-rays [5]. They also consider clinical findings such as tenderness and pain history. However, correlating symptoms with pulpal status is often challenging in pediatric patients [6]. In some uncertain cases, caries removal method is employed in that removal of the carious lesion is done layer-by-layer and in small amounts preserves the pulp vitality and intactness in the case if the caries has not reached the pulp yet [7]. Making this decision requires substantial knowledge and experience.

Furthermore, in borderline cases where it is uncertain whether pulp treatment is necessary, obtaining informed consent from parents becomes challenging. Clinicians must consider parents' preferences, the family's ability to manage dental care costs, and the uncertainty surrounding the need for pulp treatment, which add to the complexity of treatment planning. A reliable diagnostic tool would streamline the treatment process, making it easier to obtain informed consent and provide a more accurate cost estimate to parents. This would benefit not only students in dental school clinics but also early-career dentists.

Artificial intelligence (AI), particularly deep learning (DL) with convolutional neural networks (CNN), is rapidly advancing in dental research. Especially in imaging, the use of DL with CNNs to process various types of images has been actively researched and has shown promising performance [8,9]. Recently successful AI models have emerged for caries detection [10], dental age estimation [11–13], sex estimation from teeth [10], working length

determination, and morphology detection [14], and assessing endodontic treatment difficulty [15].

The process of classification in a CNN model involves: 1) Convolutional layers (the first step) to extract features such as gradients or edges from the input image using the mathematical transformations, 2) Non-linear activation functions, which are placed between any two layers and guide the input signals into output signals required for the NN to act, 3) Pooling layer, which reduces the number of parameters to learn and the amount of computation to summarize the features generated by the convolution layer, 4) Fully connected layers that are responsible for the interpretation of the feature representations learned by preceding layers [16]. YOLO is an algorithm employing CNN principles above, designed to precisely identify objects in real-time [17].

The aims of this study were: (1) to develop an ML model to diagnose the risk of pulp involvement in bitewing images of carious primary molars, and (2) to test the diagnostic performance of the developed model in a new set of samples.

## Methods and sample

### Ethics statement

This study was conducted in accordance with the ethical standards of the Declaration of Helsinki. Ethical approval was obtained from the Research Ethics Board of the University of Alberta (Approval Number: Pro00121109).

### Study population

This study was approved by the Health Research Ethics Board University of Alberta (HREB-Pro00121109). Clinical charts of 900 pediatric patients treated by Year 3 and Year 4 dental students between January 2018 and December 2022 at the University of Alberta Kaye Edmonton Clinic were screened. A sample of patients with carious primary molars was identified. Bitewing radiographs of the treated carious primary molars were examined, along with clinical data including post-treatment diagnoses of pulp involvement condition. Data was obtained from the axiUm database of the pediatric dentistry clinic. Samples with missing radiographic or clinical data, and low-quality images, such as those with cone-cuts, overlaps, or other artifacts, were excluded.

### Data preprocessing

Preprocessing steps involved image standardization to a consistent resolution, size (280X280 pixels), and format, which was automatically adjusted to 288 to ensure compatibility with the model's stride. The images were then normalized to compensate for under- and overexposure. In the preprocessing stage, the bitewing radiographs were cropped to display only one tooth per image in the optimal position. If multiple teeth in the same patient were treated during a session, the bitewing images were cropped to create individual images for each treated tooth.

### Sample labeling

Clinical data from patient charts were used as ground truth to label bitewing radiographs for training the proposed AI model. The images were labeled as "pulp involved", for caries with definitive pulp involvement, and "pulp not involved", for caries without definitive pulp involvement (teeth with caries only), according to the real final clinical situation of pulp involvement as registered in the patients' dental chart records and received treatment.

The two datasets, which comprised radiographs of lower and upper primary molars were labeled. Data extraction was repeated twice on 30 images to ensure accuracy. The datasets were then split into training, validation, and test sets with a ratio of approximately 70:20:10, Data augmentation was done before splitting ensuring no overlap between the sets. This was done using techniques such as rotation, scaling, horizontal flipping, and translation to increase data variability. The final cohort of radiographs included 688 images for training and 253 images for validation of upper primary molars, and 607 images for training and 238 images for validation of lower primary molars.

## Model architecture

The YOLOv8m-cls model architecture included multiple convolutional layers followed by classification layers. The training utilized the AdamW optimizer with an auto-determined learning rate of 0.0001 and momentum of 0.9, aiming to minimize the classification loss function. The training process consisted of 40 epochs for both datasets, with early stopping based on validation loss to prevent overfitting. A batch size of 16 was used, and Automatic Mixed Precision (AMP) was employed to accelerate training by utilizing mixed precision arithmetic while maintaining numerical stability. Pretrained weights were utilized, with 228 out of 230 items successfully transferred to facilitate better starting performance and improved convergence. The entire training process, including the results and model architecture, was logged using TensorBoard, enabling detailed visualization and analysis of the training and validation phases.

Model performance was evaluated using top-1 accuracy metrics on the validation sets after each epoch. The final trained models' performances were assessed on the set of new unseen test images (24 lower primary molars with pulp involvement and 32 lower primary molars without pulp involvement for the lower teeth model, and 31 upper primary molars with pulp involvement and 31 upper primary molars without pulp involvement for the upper teeth model) to evaluate its generalization capability in classifying primary tooth radiographs based on pulp involvement.

## Visualization of model diagnosis

Gradient-weighted Class Activation Mapping (Grad-CAM) was used to visualize the regions of an image that were most focused on by the model during its decision-making process and to improves model transparency. Grad-CAM is a visualization technique that generates heatmaps to highlight the most influential areas in a neural network's predictions. By computing the gradients of the target class with respect to the feature maps, Grad-CAM produces an activation map that overlays onto the original image, providing insight into the model's focus areas [18].

## Statistical analysis

Classification accuracy measures were used to evaluate outcomes from the test image sets. In ML different evaluation metrics are applied according to the type of problem. Accuracy, precision, recall, and the F1-score are used for classification tasks. As this study was based on a classification task, the evaluation criteria of accuracy, precision, recall, and the F1-score were used to evaluate the classification performance of the proposed model. A confusion matrix was used to calculate these values [19]. The confusion matrix has true positive (TP), true negative (TN), false positive (FP), and false negative (FN) values. The equations for accuracy, recall, precision, and the F1-score, which are performance evaluation metrics, are provided below:

$$Accuracy = (TP + TN) / TP + TN + FP + FN$$

$$Recall = TP/(TP + FN)$$

$$Precision = TP/(TP + FP)$$

$$F1\text{-}score = 2 \ X (precision \ X \ recall)/(precision + recall)$$

## Results

A sample of patients with carious primary molar teeth was selected, ranging in age from 3.8 to 11.7 years, with a median age of 7.8 years. Their clinical charts included 482 carious primary molars: 239 lower and 243 upper primary teeth. Of the 239 lower teeth, 124 received pulp treatment, while 115 were restored without pulp treatment. Among the 243 upper teeth, 120 had pulp involvement, and 123 received only restorations. The model performance for the upper teeth showed a top-1 accuracy of 78.7% and a top-5 accuracy of 1, while the results for the lower teeth achieved a top-1 accuracy of 87.8% and a top-5 accuracy of 1 (Fig 1-A and B).

Table 1 represents the model performance on the classification of primary molars in the validation datasets for both upper and lower primary molars. The model showed higher accuracy in classifying the lower primary teeth compared to the upper primary molars.

Table 2 depicts the performance of the model on classifying two new sets of test images for upper and lower teeth. The F1-Score value, which is usually used to judge the overall performance of the model and is defined as the average of precision and recall values, showed that our proposed models performed highly accurately in classifying the involvement of pulp in

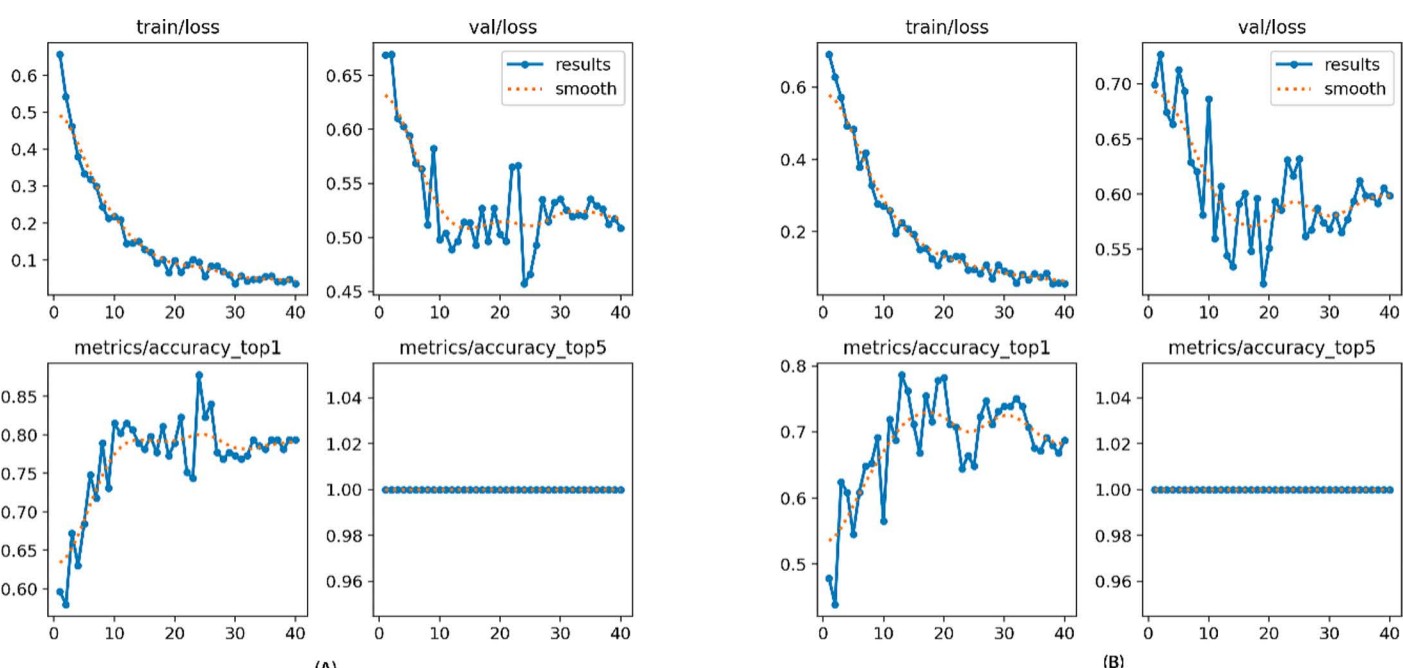

**Fig 1. Yolo-v8 classification model results on upper (A) and lower (B) primary molar.**

**Table 1. Performance of the model on validation datasets for classifying the primary molars based on the pulp involvement.**

| Metric | Upper primary molars | Lower primary molars |
|---|---|---|
| Precision | Pulp involved: 0.75<br>Pulp not involved: 0.84 | Pulp involved: 0.96<br>Pulp not involved: 0.97 |
| Recall | Pulp involved: 0.86<br>Pulp not involved: 0.72 | Pulp involved: 0.79<br>Pulp not involved: 0.82 |
| Accuracy | Pulp involved: 0.79<br>Pulp not involved: 0.79 | Pulp involved: 0.88<br>Pulp not involved: 0.88 |
| F1-score | Pulp involved: 0.80<br>Pulp not involved: 0.77 | Pulp involved: 0.87<br>Pulp not involved: 0.89 |

**Table 2. Performance of the model on a new set of images for classifying the primary molars based on the pulp involvement.**

| Metric | Upper primary molars | Lower primary molars |
|---|---|---|
| Precision | Pulp involved: 0.74<br>Pulp not involved: 0.79 | Pulp involved: 0.88<br>Pulp not involved: 0.83 |
| Recall | Pulp involved: 0.81<br>Pulp not involved: 0.71 | Pulp involved: 0.88<br>Pulp not involved: 0.83 |
| Accuracy | Pulp involved: 0.74<br>Pulp not involved: 0.79 | Pulp involved: 0.88<br>Pulp not involved: 0.83 |
| F1-score | Pulp involved: 0.77<br>Pulp not involved: 0.75 | Pulp involved: 0.89<br>Pulp not involved: 0.83 |

primary teeth using bitewing radiographs. Similarly, the accuracy of the model in the correct classification of teeth with and without pulp involvement was greater in lower primary molars.

Grad-CAM visualizations are presented in Fig 2, these heat maps were generated to illustrate the model's focus areas during classification for both positive (pulp involved) and negative (pulp not involved) samples.

## Discussion

In this study, we developed a CNN model using YOLOv8m-cls for predicting pulp involvement on bitewing radiographs in carious primary molars. Our model demonstrated high classification accuracy of over 75% for the upper and over 80% for the lower teeth and showed the potential to significantly assist clinicians in diagnosing pulp involvement in clinical settings.

In the present study, we used bitewing images of primary teeth and trained a YOLO-v8 classification model to classify teeth into two distinct groups based on the involvement of the pulp. The annotating step was skipped in the proposed model, which resulted in a more time-efficient image pre-processing. In the literature, older versions of YOLO have been widely used in the field of dental radiology to detect mandibular fractures in panoramic radiographs [20], primary and permanent tooth detection on pediatric dental radiographs [21], detection of cysts and tumors of the jaw [22], and detection of impacted mandibular third molar teeth [23]. However, new versions of YOLO-v8, released in January 2023, demonstrate superior performance for throughput and computational load requirements [24], and provide a network architecture that requires lower computing and training requirements, hence providing a more effective feature integration method, more accurate object detection performance, a more robust loss function, and an increased label assignment and model training efficiency [25,26].

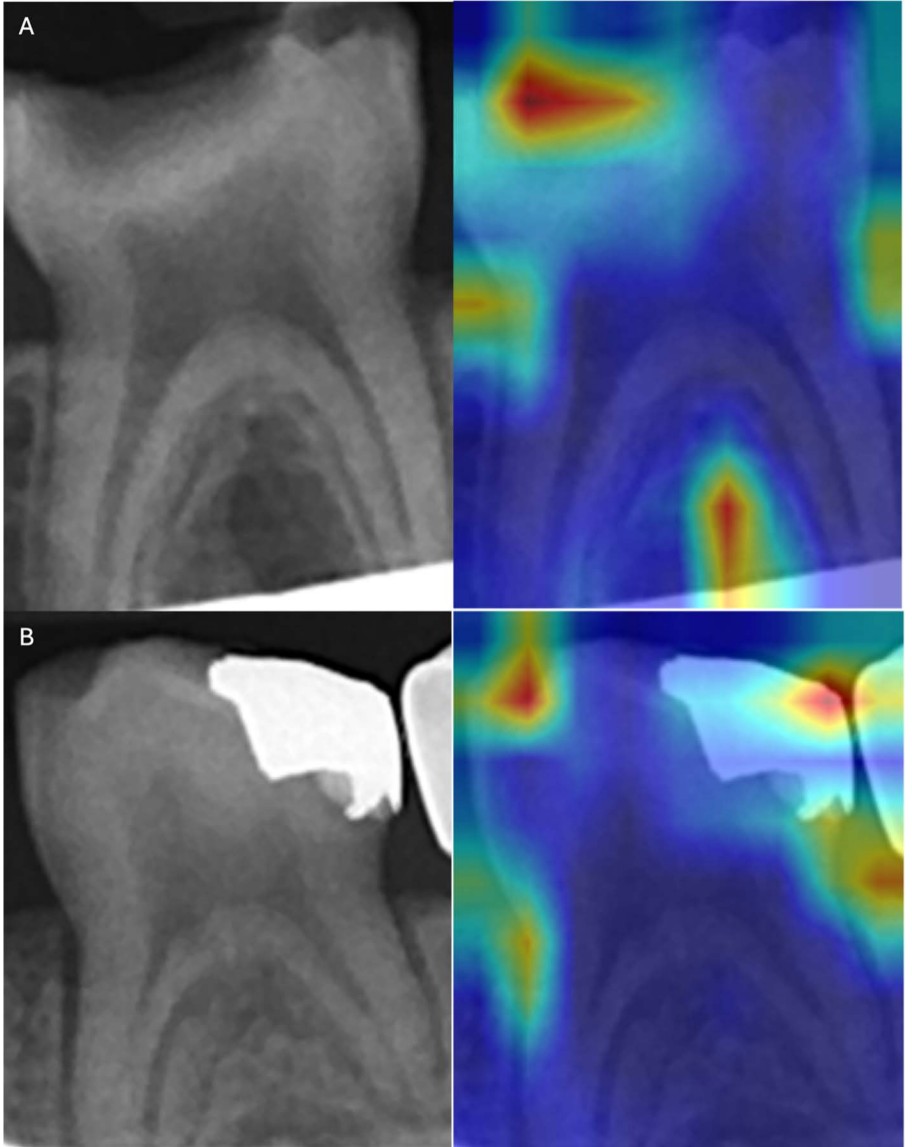

**Fig 2. Grad-CAM visualizations illustrate the model focus areas during classification for positive (A) (pulp involved) and negative (B) (pulp not involved) samples.**

The final phase of most ML studies is diagnostic performance or accuracy testing. In this phase, the findings of ML model are validated against a gold standard defined as an error-free reference standard [27]. In the present study, the post-treatment diagnosis was used as the ground truth while most ML studies in dental and medical domains rely on diagnoses from expert panels, which can exhibit errors due to the subjective nature of the task, individual expertise, biases towards certain diagnoses, among other factors. This introduces limitations regarding the certainty of the ground truth labels. It is crucial to recognize that while expert panel labeling offers necessary reference points for model training and evaluation, it may not always reflect the true ground truth [28]. Instead, we validated the model against the actual final diagnoses recorded in the patients' clinical charts to determine pulp involvement. The images were labeled based on this information rather than relying solely on the diagnosis of

dentists. This approach allowed the developed model to exceed the accuracy of experienced dentists, approaching maximum accuracy.

In the present study, if validated against the diagnoses of one or more experts, the accuracy of the developed model would likely match the expert level. It is acknowledged that for any individual specialist, accuracy is rarely 100%, and there can be instances where a dentist unexpectedly encounters pulp involvement when removing a carious lesion.

In a recent study, Ma et. al. developed a model to classify deep carious primary molars of patients 3–8 years old into two groups: 1) The failure group defined as teeth that required non-vital pulp therapy or extraction during the next 18-month observation period, and 2) The success group defined as teeth that remained clinically and radiographically free of pathology during the observation period. They labeled their samples based on the actual clinical outcomes rather than relying solely on the expert panel diagnosis. They utilized 348 samples of periapical radiographs of primary teeth, achieving 90% accuracy in predicting the prognosis. The outcomes included whether the teeth received non-vital pulp therapy, were extracted, or remained pathology-free during observation [29]. The methodologies of the two studies are comparable, with both demonstrating that utilizing ML models is promising for assessing pulp conditions in primary molars. However, there are differences as well. Ma et al. labeled samples as either needing pulp treatment or extraction within the same group, while our study excluded extracted samples, aiming to develop a model specifically for diagnosing teeth in need of pulp treatment. Their study included 65 teeth in the "failure" group and 357 in the "success" group. Another difference is related to the time interval between the x-ray taken and the treatment performed. They selected a period of up to 18 months, whereas we limited it to 6 months, excluding any x-rays taken more than 6 months before the treatment session.

In a recent study, Wang et al. evaluated ML-based predictions of pulp exposure in permanent teeth using periapical radiographic images. Their methodology closely resembled ours, particularly in the selection of reference standard and case-labeling processes. However, their study focused on permanent teeth and periapical images, whereas ours concentrated on primary teeth and bitewing radiographs. Additionally, while Wang et al. employed three ML models (VGG, ResNet, and DenseNet), we adapted the YOLOv8 architecture for classification. They trained the models on 126 samples (56 with no exposure and 70 with exposure in posterior teeth). The reference standard in this study was clinical pulp exposure observed after cavity preparation. They tested the developed models on a separate set of 40 samples (24 without exposure and 16 with exposure). DenseNet achieved the best performance in their study (AUC = 0.97), followed by ResNet (AUC = 0.89), VGG (AUC = 0.78), and senior dentist (AUC = 0.87). Their reported accuracy levels for the three trained models ranged between 75% and 85%, which are comparable to ours (78.7%–87%) [30]. Ramezanzade et al. also compared the effect of providing AI-based radiographic information versus standard radiographic and clinical information on dental students' ability to predict pulp exposure. While their study used bitewing radiographs similar to ours, the primary goal was to assess the impact of AI assistance through the application of convolutional neural networks (CNN) based on ResNet-50 architecture on students' diagnostic capabilitieson students' diagnostic capabilities. Their findings showed that AI outperformed the students, the participants only benefited "slightly" when provided with AI predictions. Notably, the authors emphasized the need for more explainable AI systems to address real-world clinical complexities, including factors such as provider experience and patient expectations [31].

We used a comparable number of "positive" and "negative" cases in test sets. It is known that diagnostic accuracy is affected by disease prevalence [32]. Disease prevalence in a population impacts positive and negative predictive values. With high positive cases in the test set, a test will be reported as more effective at 'ruling in' the positive condition and less effective at

'ruling it out [33]. Consequently, having a representative and comparable prevalence of "positive" and "negative" conditions will assure researchers about the validity and the generalizability of the results. This point is reflected in the comparable levels of accuracy in the preditction of the "pulp involved" and "pulp not involved" categories in our study (0.74 vs 0.79 for upper molars, and 0.88 vs 0.83 for lower molars).

In the present study, a small difference was observed in the precision achieved for upper molars compared to lower molars. While the clinical significance of this difference is uncertain, it can likely be attributed to the more complicated anatomy of pulp chambers in the upper molars, due to the complex anatomy of roots and canals compared to lower molars, as well as the fact that upper molars typically have three roots versus the two roots in lower molars.

This study has some limitations that need to be acknowledged. First, cases retrieved from the University pediatric clinic were treated by undergraduate dental students under the supervision of experienced instructors. This may have introduced the possibility of over-cavity preparation and dentine removal in some carious teeth, leading to a higher number of exposed pulps compared to ideal conditions. This could result in a bias towards predicting more pulp involvement in borderline cases. In addition, we cropped the included tooth from the bitewing images for training the ML model. Consequently, this ML model is designed to accurately analyse single tooth images rather than entire bitewing images. For future studies, a better approach would be to use whole bitewing images to train new models, enabling AI to be flexibly applied to entire images. However, this approach requires additional training steps to differentiate between intact teeth, buds of unerupted permanent teeth, and erupting permanent teeth visible in the images. Third, we used images of the carious teeth to develop the model and integrated clinical examination data along with patient-related factors such as age, eruption stage of underlying permanent teeth, estimated time to eruption of successors, and other relevant (influential) treatment planning variables. This comprehensive approach has the potential to significantly enhance the model's accuracy. However, the ease of use in clinical practice should also be considered, as an increase in the number of input factors might make the model more complex and time-consuming to use. Finally, we did not create heat-map images to identify the focus areas of the trained model in detecting pulp involvement, which means that the principles by which this AI model diagnoses the possibility of pulp involvement are still unknown. As this is the first AI study in the field, our primary objective was to evaluate the feasibility of this approach. Therefore, future projects may focus on creating heat maps and uncovering potential hidden features in this context.

Refined models using this method could enhance the accuracy of diagnoses and treatment plans in clinical practice for pediatric dentists. This improvement would aid in providing more reliable cost estimates for treatments and obtaining precise informed consent. Additionally, it could streamline patient scheduling by better estimating the need for pulp treatment and help general dentists distinguish between manageable and complex cases, prompting appropriate referrals. Moreover, the model serves as a valuable training tool for dental students, allowing them to compare their diagnostic assessments with AI predictions and enhancing their clinical skills. The rapid development of smartphones presents an opportunity to streamline dental practice through mobile apps. Developing an app based on this model would offer convenient access, akin to commercially available apps like WebCeph for orthodontic diagnosis.

## Conclusion

This study successfully developed a convolutional neural network (CNN) model, specifically using YOLOv8m-cls, for diagnosing pulp involvement in primary teeth from bitewing radiographs, achieving high classification accuracy. The model demonstrated over 75% accuracy

for both upper and lower primary molars, with superior results for lower molars. Future research should focus on training models with whole bitewing images and incorporating additional clinical variables to further improve accuracy. Despite some limitations, this study lays a solid foundation for AI applications in pediatric dentistry, empowering clinicians to make informed decisions, enhance patient care processes, and advance dental education. The potential development of a mobile app underscores the transformative impact of AI on dental diagnostics and treatment planning, particularly in pediatric practice.

## Author contributions

**Conceptualization:** Aydin Sohrabi, Masoud MiriMoghaddam, Maryam Amin.

**Data curation:** Aydin Sohrabi, Masoud MiriMoghaddam.

**Formal analysis:** Nazila Ameli.

**Funding acquisition:** Aydin Sohrabi, Maryam Amin.

**Investigation:** Aydin Sohrabi.

**Methodology:** Aydin Sohrabi, Nazila Ameli, Maryam Amin.

**Validation:** Yuli Berlin-Broner, Hollis Lai.

**Writing – original draft:** Aydin Sohrabi, Nazila Ameli.

**Writing – review & editing:** Aydin Sohrabi, Nazila Ameli, Masoud MiriMoghaddam, Yuli Berlin-Broner, Hollis Lai, Maryam Amin.

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
