## [Decision Letter · Decision Letter 0]

23 Dec 2024

PDIG-D-24-00451Deep Learning-Based Assessment of Pulp Involvement in Primary Molars Using YOLO v8PLOS Digital Health Dear Dr. Amin, Thank you for submitting your manuscript to PLOS Digital Health. After careful consideration, we feel that it has merit but does not fully meet PLOS Digital Health's publication criteria as it currently stands. Therefore, we invite you to submit a revised version of the manuscript that addresses the points raised during the review process. Please submit your revised manuscript within 60 days Feb 21 2025 11:59PM. If you will need more time than this to complete your revisions, please reply to this message or contact the journal office at digitalhealth@plos.org. Please include the following items when submitting your revised manuscript:* A rebuttal letter that responds to each point raised by the editor and reviewer(s). You should upload this letter as a separate file labeled 'Response to Reviewers '. This file does not need to include responses to any formatting updates and technical items listed in the 'Journal Requirements' section below.* A marked-up copy of your manuscript that highlights changes made to the original version. You should upload this as a separate file labeled 'Revised Manuscript with Track Changes '.* An unmarked version of your revised paper without tracked changes. You should upload this as a separate file labeled 'Manuscript '. If you would like to make changes to your financial disclosure, competing interests statement, or data availability statement, please make these updates within the submission form at the time of resubmission. Guidelines for resubmitting your figure files are available below the reviewer comments at the end of this letter. We look forward to receiving your revised manuscript. Kind regards, Ismini LourentzouSection EditorPLOS Digital Health Ismini LourentzouSection EditorPLOS Digital Health Leo Anthony CeliEditor-in-ChiefPLOS Digital Healthorcid.org/0000-0001-6712-6626 **Journal Requirements:**

1. We ask that a manuscript source file is provided at Revision. Please upload your manuscript file as a .doc, .docx, .rtf or .tex.

2. In the online submission form, you indicated that The datasets used and analyzed during the current study are available from the corresponding author upon reasonable request. All data generated or analyzed during this study are included in this published article.

3. Uploaded as supplementary information.

**Additional Editor Comments (if provided):****Reviewers' Comments:** Reviewer's Responses to Questions

**Comments to the Author**

1. Does this manuscript meet PLOS Digital Health’s publication criteria ? Is the manuscript technically sound, and do the data support the conclusions? The manuscript must describe methodologically and ethically rigorous research with conclusions that are appropriately drawn based on the data presented.

Reviewer #1: Partly

Reviewer #2: Yes

Reviewer #3: Yes

2. Has the statistical analysis been performed appropriately and rigorously?

Reviewer #1: Yes

Reviewer #2: Yes

Reviewer #3: Yes

3. Have the authors made all data underlying the findings in their manuscript fully available (please refer to the Data Availability Statement at the start of the manuscript PDF file)?

Reviewer #1: No

Reviewer #2: Yes

Reviewer #3: Yes

4. Is the manuscript presented in an intelligible fashion and written in standard English?

Reviewer #1: Yes

Reviewer #2: Yes

Reviewer #3: Yes

5. Review Comments to the Author

Reviewer #1: Comments

Main Claims and Significance

The manuscript claims to have developed a machine learning model using the YOLOv8 architecture to diagnose pulp involvement in primary molars through bitewing radiographs. This is significant for pediatric dentistry as it offers a potentially more accurate and efficient method for assessing dental caries, which is a prevalent issue among children .

Novelty and Contextual Placement

The use of the YOLOv8 model is relatively novel in this specific application, as previous studies have utilized older versions of YOLO for different dental diagnostics. The manuscript should better contextualize its claims by comparing them with existing literature, particularly studies that have used similar machine learning approaches in dental radiology for assessing pulp involvement. This would help in establishing the originality and contribution of the current study.

Data and Analysis Support

The data and analyses appear to support the claims, with the model achieving notable accuracy rates for both upper and lower primary molars. However, the study could benefit from additional validation, such as comparisons with diagnoses from dentists.

Recommendations for Improvement

As per the Tripod+ai guideline for reporting clinical prediction models using regression or machine learning methods, below are my recommendations:

1. Comparison with reference standard model/other models: The study can benefit by including either:

a. an objective comparison with an existing data science model predicting the same outcome.

b. a clinical score used in clinical settings objectively using a numeric value or a metric (if outcome of score and algorithm is same).

c. an objective comparison with results obtained by a group of dentists.

d. an objective comparison based on clinical experience/judgement.

2. Reproducibility: The manuscript should provide more detailed descriptions of the methodology, including data preprocessing and model training steps, to ensure reproducibility. A GitHub repository or at least supplementary material should suffice.

3. Calibration performance of the model: Calibration performance metrics like calibration plot or Brier score should be evaluated.

4. Reporting of parameter estimates: We can also present the initial and final values of weights and biases for each layer of the CNN. This could be done in a tabular format or as part of an appendix to avoid cluttering the main text. Also, a comprehensive description of the model architecture, specifying the number of layers, types of layers (e.g., convolutional, fully connected), and activation functions used would be helpful.

5. Model interpretability: Incorporating heat maps could provide insights into the model's decision-making process, enhancing transparency and understanding of the AI's focus areas.

Reviewer #2: The topic was interesting and there is certainly value in the area of dental education, especially in a movement towards selective caries removal and vital pulp therapy. A couple of comments:

(1) The title and thrust of the manuscript implies that radiographic findings alone dictate decision-making in treatments of primary pulp tissue. While the authors do mention the importance of clinical findings when generate treatment plans, it is unclear how this deep learning-based assessment will account for the histologic status of the pulp tissue and the clinical findings. Consider rewording the title.

(2) The authors do a good job understanding and describing the limitations of the study. There would be value in a follow-up study that looks at the long-term outcomes of the teeth that were treated conservatively, to understand if the model can reliably predict long-term outcomes of various treatment modalities.

Reviewer #3: In diagnosing the relationship between caries in primary molars and the pulp in children using the bitewing method, if AI support constructed in this study can be obtained, it may lead to an improvement in prognosis. However, one point of caution in building the AI is the labeling accuracy during dataset construction. Was the labeling done by pediatric dentistry specialists? How was the judgment made by referring to medical information, electronic medical records, and clinical information? What were the exclusion criteria for each case? Furthermore, is there any variation in human judgment? Medical information, electronic medical records, and clinical information may not always accurately reflect phenomena or prognosis; sometimes, they are recorded for convenience in calculating treatment costs. From a dental perspective, referring to medical information, electronic medical records, and clinical information, the influence of the initial diagnosis at the time of the first consultation cannot be ignored. In other words, the diagnosis of caries is not standardized by dentists, and in cases where there is no problem with the pulp, pulp treatment might have been unnecessarily performed, without any backup support. Therefore, it is necessary to address the issue of reliability concerning the evidence obtained from the retrospective study in this research.

---

## [Decision Letter · Decision Letter 1]

6 Mar 2025

Deep Learning-Based Assessment of Pulp Involvement in Primary Molars Using YOLO v8

PDIG-D-24-00451R1

Dear Dr. Amin,

We are pleased to inform you that your manuscript 'Deep Learning-Based Assessment of Pulp Involvement in Primary Molars Using YOLO v8' has been provisionally accepted for publication in PLOS Digital Health.

Best regards,

Ismini Lourentzou

Section Editor

PLOS Digital Health

**Additional Editor Comments (if provided):**

**Reviewer Comments (if any, and for reference):**

Reviewer's Responses to Questions

**Comments to the Author**

1. If the authors have adequately addressed your comments raised in a previous round of review and you feel that this manuscript is now acceptable for publication, you may indicate that here to bypass the “Comments to the Author” section, enter your conflict of interest statement in the “Confidential to Editor” section, and submit your "Accept" recommendation.

Reviewer #1: All comments have been addressed

Reviewer #2: All comments have been addressed

Reviewer #3: All comments have been addressed

2. Does this manuscript meet PLOS Digital Health’s publication criteria ? Is the manuscript technically sound, and do the data support the conclusions? The manuscript must describe methodologically and ethically rigorous research with conclusions that are appropriately drawn based on the data presented.

Reviewer #1: Yes

Reviewer #2: Yes

Reviewer #3: Partly

3. Has the statistical analysis been performed appropriately and rigorously?

Reviewer #1: Yes

Reviewer #2: I don't know

Reviewer #3: N/A

4. Have the authors made all data underlying the findings in their manuscript fully available (please refer to the Data Availability Statement at the start of the manuscript PDF file)?

Reviewer #1: No

Reviewer #2: Yes

Reviewer #3: No

5. Is the manuscript presented in an intelligible fashion and written in standard English?

Reviewer #1: Yes

Reviewer #2: Yes

Reviewer #3: Yes

6. Review Comments to the Author

Reviewer #1: The manuscript now addresses the potential shortcomings identified earlier and although there is further scope in improvement- it seems to meet the criteria for publication.

Reviewer #2: (No Response)

Reviewer #3: In addition to top-1, top-5 was included as an evaluation metric, with the results reported as 1. However, the Material and Methods section suggests that two types of labeling were conducted. Given this, what was the necessity of incorporating top-5 in the evaluation?
